# Reverse Logistics Network Design for Medical Waste Disposal under the Scenario of Uncertain Proposal Demand

Lan Zhu , Tao Ding and Zhuofan Liu *

School of Modern Post, Xi'an University of Posts and Telecommunications, Xi'an 710061, China;
zhulan@xupt.edu.cn (L.Z.); dingtao@stu.xupt.edu.cn (T.D.)
* Correspondence: zhuofan.liu@xupt.edu.cn

**Abstract:** With the development of the healthcare industry, the demand for medical services and protective equipment is boosted, causing the generation rate of infectious medical waste to increase rapidly. Therefore, it is of utmost importance for decision makers to effectively predict the potential risks and propose corresponding solutions. This paper investigates the reverse logistics network optimization for medical waste under the conditions of an uncertain proposal demand. Firstly, a prediction model of medical waste based on the SEIR epidemiological dynamics is constructed, in which both routine and public health emergency scenarios are simultaneously considered. Secondly, a bi-objective location-routing optimization model for a medical waste reverse logistics network is proposed, by simultaneously optimizing the total economic cost and potential risk throughout the entire logistics process. Subsequently, an NSGA-II algorithm is designed for a model solution in response to the model's characteristics. The epidemiological dynamics-based prediction model is validated by the real case to be scientifically effective in predicting the amount of generated medical waste with a mean absolute percentage error (MAPE) of 18.08%. The constructed reverse logistics network model and the NSGA II algorithm provide a medical waste process center location, transportation routing, and vehicle selection solutions for both routine and emergency public health cases of Xi'an city with large, medium, and small scales. The above results indicate that the research scheme proposed in this paper could significantly reduce the medical waste logistics-related risks and costs and provide decision makers with more safe and reliable logistical solutions.

**Keywords:** medical waste; reverse logistics network; SEIR model; location-routing problem; NSGA-II

## 1. Introduction

In recent years, with the acceleration of urbanization and the frequent outbreaks of new infectious diseases, people's attention to the issue of medical waste disposal has gradually risen, and the standardized disposal of medical waste has become an important issue of social concern. Medical waste refers to waste generated by healthcare institutions in medical, preventive, healthcare, and other related activities, possessing direct or indirect infectious, toxic, and other hazardous characteristics. As a specialized type of industrial waste, it carries high spatial pollution and potential transmission risks [1]. Due to these reasons, neglecting risk control in logistics processes such as the recovery, treatment, and transportation of medical waste will result in serious harm to nearby residents and the surrounding environment [2]. Particularly, in the event of a public health emergency, the untimely transport or disposal of medical waste carrying pathogens can pose a significant threat to epidemic prevention and control. Therefore, the construction of a well-functioning and flexible reverse logistics network for medical waste is an urgent and practical problem that needs to be addressed.

Current research on reverse logistics networks [3] optimization for medical waste typically encompasses decision problems such as facility location and vehicle routing. From a strategic perspective, the selection of key nodes, such as temporary storage points

and processing centers, directly influences the layout of the logistics network, thereby impacting disposal efficiency and risk control. From an operational perspective, the choice of routes for transportation vehicles significantly determines transportation efficiency and is crucial for the safety of residents and the environment along the way. In previous studies, these two decision problems related to location and routing are often considered separately, with the locations of critical facilities being determined first, followed by the planning of transportation routes [4]. However, considering that the selection of such avoidance facilities and the transportation routes for their specific waste is relatively fixed, integrated decision-making can lead to a globally optimal planning result for the system [5]. Many experts study the integrated location-routing problem in medical waste logistics from various perspectives such as economic, environmental, and multiple time periods. These include location-routing models for medical waste [6], dual-level equilibrium location-allocation optimization models [4], and multi-objective programming models [7] to comprehensively address decisions related to the location selection and transportation of medical waste disposal facilities. However, since medical waste contains tissues containing pathogenic microorganisms such as a patient's vomit and body fluids, as well as disposable instruments, needles, syringes, surgical blades, syringes, and other wastes used in operating theatres, emergency rooms, and injection rooms, etc., which have a certain degree of viral transmissibility, and these wastes are prone to cause environmental contamination and the spread of pathogens in the reverse logistic activities, the traditional optimization methods are only concerned with the minimization of the costs, which are not able to satisfy the current demand for the disposal of medical wastes.

In view of this, many experts have approached the optimization of reverse logistics networks for medical waste from the perspective of risk control, aiming to minimize the probability or severity of accidents involving leakage during transportation or on-site incidents that could cause harm to the surrounding population, property, and the environment [8]. For instance, some experts considered the impact of transportation time on risk [9], and constructed a medical waste transportation risk measurement model, in which the risk-bearing capacity of the network nodes is expressed as the product of the transportation volume and corresponding transportation time. Other experts considered the infectivity of viruses carried by medical waste on the transportation risk and expressed the risk as the product of the infection probability of medical waste, the quantity of transported medical waste, and the transportation distance [10]. They conducted a sensitivity analysis on the infection probability to observe its impact on the results. However, the aforementioned studies only subjectively defined the probability of risk, without considering the influence of the quantity of medical waste on the risk probability. Therefore, in this paper, we start from the probability of risk occurrence and construct a risk probability calculation function influenced by the quantity of medical waste generated to quantify the impact of different amounts of medical waste in the environment on the risk probability.

Furthermore, the accurate prediction of medical waste disposal demand is a crucial prerequisite for the layout of reverse logistics facilities and the allocation of the disposal capacity. Currently, the most common approach to predicting medical waste generation is to analyze influencing factors during its development process, and extrapolating future states based on historical data. For example, the trend extrapolation method [11] uses the volume of medical waste generated over a certain period as a basis and analyzes its changing patterns to predict future generation. Though this method is simple and practical, it ignores the relationship between past and future medical waste generation. Regression analysis [12], on the other hand, focuses on the factors related to medical waste generation and analyzes the correlation of each factor with the least squares method. Some experts treat the volume of medical waste generation as an uncertain parameter in optimization models, dealing with the uncertainty of waste generation through methods like fuzzy chance-constrained programming [13] and grey prediction models [14]. These approaches construct equivalent deterministic expressions for uncertain waste generation and calculate the volume of medical waste. Additionally, considering the volatility of medical waste generation during

a public health emergency, some experts also use the number of cases to characterize the amount of medical waste generated [15]. There are also studies suggesting an approach that simulates the trend of epidemic development, utilizing simulation methods to elucidate the relationship between medical waste generation and the scale of the outbreak, such as infectious disease dynamics and artificial neural networks [16].

Based on the above analysis, this paper comprehensively investigates the optimization problem of reverse logistics networks for medical waste. The main research contents are as follows. Firstly, the integrated optimization problem of site selection for medical waste processing and transportation route decisions considering risk factors is studied. A multi-objective location routing problem (MO-LRP) model that simultaneously considers risk and cost factors is constructed, and an improved NSGA-II algorithm is designed to solve the model. Secondly, the prediction of medical waste generation under both routine and public health emergency scenarios is conducted. Two distinct medical waste generation prediction models are designed based on the SEIR system dynamics method and the linear method, respectively. Finally, the effectiveness of the proposed model and algorithm is validated through examples from Xi'an, China, and various test cases of different scales.

## 2. Materials and Methods

### 2.1. Problem Description

The reverse logistics network for medical waste constructed in this paper is a two-tier network that consists of processing centers and collection points, as illustrated in Figure 1. Under routine circumstances, collection points typically include hospitals. In the event of a public health emergency, collection points should also encompass isolation locations for patients. Transportation vehicles depart from the processing center, and sequentially collect medical waste from each collection point and transport it back to the processing center for designated processing procedures. The main objectives of this study are to meet all proposal demands under both routine and public health emergency scenarios by making rational processing centers' location decisions and transportation routing decisions and to minimize the total logistics cost and risks at the same time.

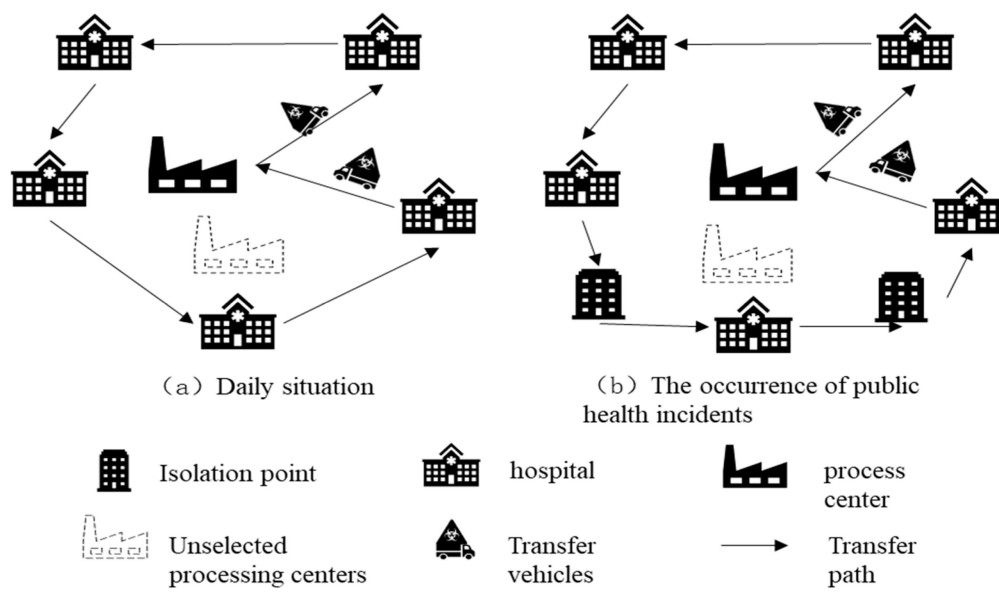

**Figure 1.** Schematic diagram of the reverse logistics network for medical waste.

In the modeling process, the following assumptions are considered. (1) The specific processing procedures for medical waste at the processing centers are not taken into account. (2) The fixed cost of transportation vehicles is independent of the transported volume, and vehicles must adhere to weight restrictions. (3) The transportation cost of transportation

vehicles is directly proportional to the distance traveled by the vehicles. (4) The time required for transportation vehicles to handle waste at collection points is not considered.

### 2.2. The Medical Waste Prediction Model Based on the SEIR Epidemiological Dynamics Method

Given that the generation of medical waste during public health emergencies is related to the development of the epidemic, this paper will, based on the epidemiological modeling and simulation methods, construct an improved SEIR dynamic prediction model. This model aims to accurately predict the uncertain generation of medical waste and specify the disposal requirements of the reverse logistics system.

In this study, it is assumed that the daily amount of medical waste generated by the hospital consists of two parts: the daily waste generation $W_{rc}$ and the waste generation $W_{yq}$ related to the occurrence of a public health emergency. In the absence of a public health emergency, only the daily waste generation is considered. The expression for the total amount of medical waste generated in the $t$-th day, $W_{(t)}$, is shown in Equation (1).

$$W_{(t)} = W_{yq} + W_{rc} \tag{1}$$

The daily generation of medical waste, $W_{rc}$, is determined by the product of the number of hospital beds ($PY_i$) and the amount of medical waste produced per bed ($\psi = 0.4$) [17], as represented in Equation (2).

$$W_{rc} = \sum_{i \in PY} PY_i \cdot \psi \tag{2}$$

Furthermore, this paper characterizes the amount of generated medical waste by the number of different populations related to the development of the epidemic process. The population is classified into four categories [18], which are the susceptible population ($S$), exposed population ($E$), infected population ($I$), and the removed population ($R$). Then, a prediction model for medical waste generation ($W_{yq}$) related to a public health emergency is constructed, as depicted in Figure 2. Here, $\beta$ and $\delta$ represent the probabilities of contact between the $E$ and $I$ population with the $S$ population, respectively. $\alpha$ represents the reciprocal of the incubation period for the virus, while $\mu$ and $\gamma$ represent the recovery and mortality rates of the $I$ population, respectively.

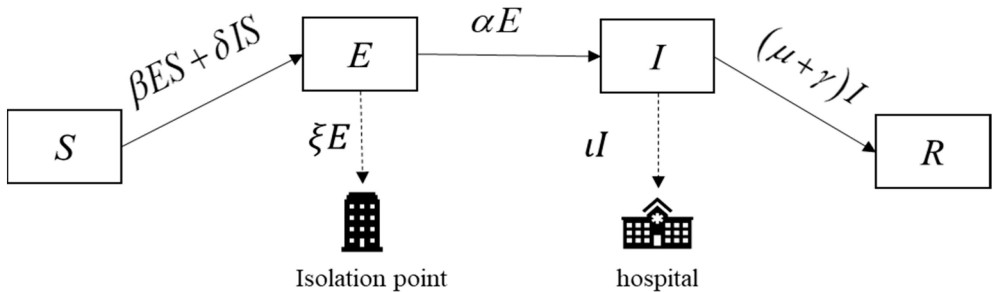

**Figure 2.** Illustration of the medical waste prediction model based on the SEIR epidemiological dynamics.

Therefore, this paper assumes that with the changing dynamics of a public health emergency, the population of the $I$ population is randomly assigned to hospitals within its jurisdiction, while the $E$ population is randomly assigned to isolation points within its jurisdiction, as shown in Figure 2. Solid arrows indicate the flow of the population with infectious viruses among different population groups, while dashed arrows represent the disposal destinations of medical waste generated by different population groups.

Equation (3) illustrates the state transition relationships among different population groups. Considering that during the sudden occurrence of a public health emergency, the $E$ population, when in contact with the $S$ population, can also cause the $S$ population to become the $E$ population. Moreover, this paper introduces an impact parameter $\beta$ of the $E$

population on the *S* population based on the traditional SEIR model to depict transition relationships more accurately.

$$\begin{cases} S_{(t)} = -\beta E_{(t)}S - \delta I_{(t)}S \\ E_{(t)} = \beta E_{(t)}S + \delta I_{(t)}S - \alpha E_{(t)} \\ I_{(t)} = \alpha E_{(t)} - (\mu + \gamma)I_{(t)} \\ R_{(t)} = (\mu + \gamma)I_{(t)} \end{cases} \tag{3}$$

When the public health emergency occurs, the predicted medical waste generation is mainly determined by the *E* population and the *I* population. Here, $\xi = 13$ and $\iota = 1.4$ [19] represent the output coefficients for the *I* and *E* population respectively, as shown in Equation (4).

$$W_{yq} = \xi \cdot I_{(t)} + \iota \cdot E_{(t)} \tag{4}$$

Therefore, the predicted equation for medical waste generation on the *t*-th day can be obtained as shown in Equation (5).

$$W_{(t)} = \sum_{i \in PY} PY_i \cdot a + \xi \cdot I_{(t)} + \iota \cdot E_{(t)} \tag{5}$$

### 2.3. Measurement of Decision Risk in the Medical Waste Logistics Network

In this paper, the risk caused by the reverse logistics of medical waste is defined as the product of the population directly affected by leakage accidents and the associated risk. This is related to both facility location decisions and routing decisions. The risk expression for the processing center ($RC_i$) in the reverse logistics network of medical waste is given by Equation (6), and the risk associated with the transportation path ($VC_{ij}$) is represented by Equation (7).

$$RC_i = sd_i \cdot \rho d_i \cdot Pd_i \cdot ad_i \tag{6}$$

$$VC_{ij} = 2r \cdot l_{ij} \cdot \rho k_{ij} \cdot Pk_{ij} \cdot ak \tag{7}$$

where $Pd_j$ is the probability (percentage) of the leakage risk for establishing a center at location *j*. $Sd_j$ is the area (square kilometers) affected by the processing center *j* in the event of leakage. $\rho d_j$ is the population density (people/square kilometer) affected by processing center *j* in the event of leakage. $\alpha d_j$ is the average per capita loss (currency/person) caused by processing center *j* in the event of leakage. $Pk_{ij}$ is the probability (percentage) of leakage risk of transfer vehicles from collection point *i* to collection point *j*. *r* is the radius (kilometers) within which the collection point *i* can cause an impact. $\rho k_{ij}$ is the population density (thousand people/square kilometer) affected by the transfer vehicle from collection point *i* to collection point *j* when leakage happens. $\alpha k$ is the average per capita loss (currency/person) caused by the transfer vehicle in the event of leakage from collection point *i* to collection point *j*.

The diagram in Figure 3 illustrates the risk impact zones for the processing centers and transportation paths.

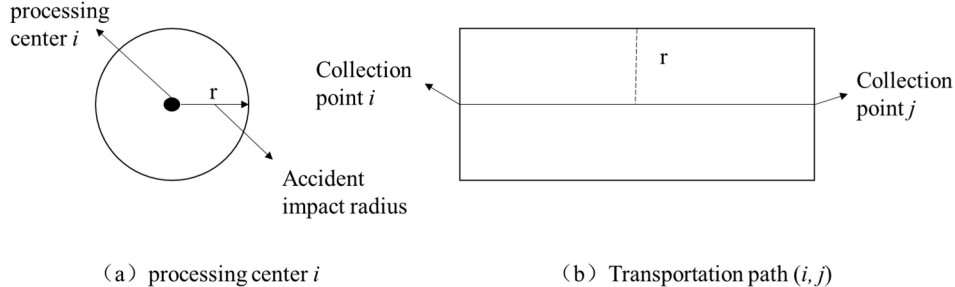

（a）processing center *i*　　　　　　　　　（b）Transportation path (*i*, *j*)

**Figure 3.** Scope of influence of processing center and transportation path.

This paper introduces the logistic growth curve model [20] to depict the variation in risk probability with the amount of medical waste generation. Before medical waste is properly treated, the growth trend of infectious pathogens it carries follows the logistic growth model as the amount of medical waste generation increases. The expression for the logistic function is shown in Equation (8).

$$P_{(t)} = \frac{KP_0 e^{rt}}{K + P_0(e^{rt} - 1)} - 1 \tag{8}$$

$P_{(t)}$ represents the probability of risk impact when the quantity of medical waste reaches a certain amount. $P_0$ is the initial amount of medical waste, $K$ is the conversion factor, and $r$ is a parameter measuring the speed of curve changes.

### 2.4. Description of the Time Window Penalty Cost

To meet the requirement of the national regulation that the medical waste must be treated on a daily basis, this paper adopts a hard time window function to restrict the travel time of vehicles on the route. Once the travel time of a vehicle exceeds the duration specified by the time window constraint, it will face substantial fines. The penalty cost function $SP_{ij}$ is defined as Equation (9).

$$SP_{ij} = M \sum_{i,j \in C} \max\{TK_{ij} - T_{\text{const}}, 0\} \tag{9}$$

### 2.5. Multi-Objective Optimization Model

The multi-objective location-routing model constructed in this paper is described by the following symbols.

Sets: $C = \{1, 2, \cdots, c\}$ is the set of collection points; $D = \{1, 2, \cdots, d\}$ is the set of processing centers; $N = C \cup D$ is the set of all nodes in the network.

Parameters: $q_i$ represents the quantity (tons) of medical waste generated at collection point $i$. $l_{ij}$ represents the distance (kilometers) from collection point $i$ to collection point $j$. $Od_j$ represents the fixed cost (CNY) of constructing processing center $j$. $Bd_j$ represents the cost (CNY/ton) of processing unit medical waste at alternative processing center $j$; $Hd_j$ represents the maximum processing capacity (tons) of processing center $j$; $Ok$ represents the fixed cost (CNY) of transfer vehicles; $Bk$ represents the transportation cost (CNY/kilometer) per unit distance traveled by transfer vehicles; $Hk$ represents the maximum payload (tons) of transfer vehicles; $V$ represents the speed (km/h) of transfer vehicles; $M$ is a positive infinity; $T_{\text{const}}$ represents the maximum allowable duration for vehicles to travel in the path; $TK_{ij}$ represents the time required for vehicles to travel in path $(i, j)$.

Decision variables: $X_j$ represents whether to construct a processing center at $j$; if a processing center is constructed, $X_j = 1$, otherwise 0; $Y_{ij}$ represents whether transfer vehicles travel from point $i$ to point $j$; if transfer vehicles travel from point $i$ to point $j$ then $Y_{ij} = 1$, otherwise 0; $Z_i^j$ represents whether collection point $i$ is serviced by processing center $j$; if collection point $i$ is serviced by processing center $j$, then $Z_i^j = 1$, otherwise 0.

Using economic cost and risk as objective functions, a medical waste reverse logistics network model is constructed as follows.

$$\min F_1 = \sum_{j \in D} Od_j X_j + \sum_{j \in D} Bd_j \sum_{i \in C} Z_i^j q_i + \sum_{i,j \in C} y_{ij} \cdot Ok + \sum_{i,j \in C} y_{ij} \cdot l_{ij} \cdot Bk + \sum_{i,j \in C} SP_{ij} \tag{10}$$

$$\min F_2 = \sum_{j \in D} X_j \cdot RC_i + \sum_{i,j \in C} Y_{ij} \cdot VC_{ij} \tag{11}$$

S.T.

$$\sum_{i \in C} Z_i^j q_i \leq Hd_j X_j, \forall j \in D \tag{12}$$

$$\sum_{i,j \in N} Y_{ij} q_i \leq Hk \tag{13}$$

$$\sum_{i,j \in C} Y_{ij} = 1 \tag{14}$$

$$\sum_{i \in C} Z_i^j = 1, \forall j \in D \tag{15}$$

$$Y_{ij} = 0, \forall i \in D, \forall j \in D \tag{16}$$

$$Z_i^j \leq X_j, \forall i \in C, \forall j \in D \tag{17}$$

$$X_j \in \{0,1\} \tag{18}$$

$$Y_{ij} \in \{0,1\} \tag{19}$$

$$Z_i^j \in \{0,1\} \tag{20}$$

Objective function (10) aims to minimize the total cost, which includes the investment cost processing cost of processing centers and the fixed cost and transportation cost of the vehicles. Objective function (11) aims to minimize the logistics risk associated with establishing processing center and transportation.

Constraint (12) represents the capacity constraint for processing center points. Constraint (13) indicates that the maximum load capacity of each vehicle should not exceed its load constraint. Constraint (14) states that each collection point should be serviced by only one vehicle. Constraint (15) specifies that each collection point should be serviced by only one processing center. Constraint (16) restricts that there are no routes between any two processing centers. Constraint (17) states that a collection point will only be serviced by a processing center if that processing center is open. Constraints (18) to (19) indicate that the decision variables are binary.

*2.6. Algorithm Design*

The medical waste reverse logistics network model designed in this paper belongs to a multi-objective combinatorial optimization model, which poses a high level of complexity. Therefore, this paper designed a fast Non-dominated Sorting Genetic Algorithm (NSGA-II) to solve the model. NSGA-II is one of the currently efficient multi-objective evolutionary algorithms, and its advantages lie in the following. Firstly, computational complexity through fast non-dominated sorting; secondly, it employs an elitist strategy to merge the parent and offspring population, enabling the next generation's population to be selected from double the space, thereby preserving excellent individuals; finally, it introduces a crowding distance comparison mechanism to ensure an even distribution of individuals across the entire Pareto domain.

The main process of the proposed NSGA-II algorithm is as follows.

Step 1: Randomly generate an initial population $P_t$ with a size of $N$, where $t = 0$;

Step 2: Apply selection, crossover, and mutation operations to the population $P_t$ to generate an offspring population $Q_t$ with a size of $N$. Merge $P_t$ and $Q_t$ into a mixed population $R_t$.

Step 3: Conduct a fast non-dominated sorting on the population $R_t$ to obtain a non-dominated solution set $\{F_1, F_2, \ldots, F_i\}$, set $P_{t+1} = \varnothing$, $i = 0$. When $P_{t+1} | + | F_i | \leq N$, and update $P_{t+1} = P_{t+1} \cup F_i$, $i = i + 1$; otherwise, calculate the crowding distance for individuals in $F_i$, sort individuals in $F_i$ in descending order based on dominance relation $< n$, and update $P_{t+1} = P_{t+1} \cup F_i [1: (N - | P_{t+1} |)]$.

Step 4: $t = t + 1$, if $t > T$ (where $T$ is the maximum number of evolution generations), terminate the algorithm; otherwise, go back to Step 2.

For the encoding of the multi-objective facility location and routing problem studied in this paper, natural number encoding is adopted. Assuming the numbers assigned to $m$ processing centers are 1, 2,..., $m$, and the numbers assigned to $n$ collection points are

*m* + 1, *m* + 2, . . ., *m* + *n*, a solution is represented by a sequence of numbers comprising the permutation of m processing centers {1, 2, . . ., *m*} and n collection points {*m* + 1, *m* + 2, . . ., *m* + *n*}. If a processing center is followed by no collection point, it signifies that the processing center is not operational. On the other hand, if a processing center is followed by a collection point, it indicates that the processing center is operational, and the frequency of occurrence of the collection point represents the number of waste recovery routes originating from that processing center.

Using the example of 3 treatment centers and 9 collection points, where 1–3 represent treatment centers and 4–12 represent collection points; processing centers that are not selected in the coding of the feasible scenarios will not be shown in the coding diagram as shown in Figure 4. Processing center 3 is operational; since processing centers 1 and 2 are not selected, they will not be shown in the coding diagram and will not be operational. There are two waste transportation routes from processing center 3: 3→6→5→11→4→7→6→3 and 3→8→12→10→9→3.

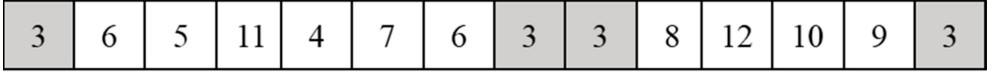

**Figure 4.** The problem encoding diagram.

## 3. Results

### 3.1. Validation of the Medical Waste Generation Forecasting Model

To validate the accuracy of the SEIR model, this paper uses the mean absolute percentage error (MAPE) between the model-calculated results, as shown in Equation (21). Here, $R_{(t)}$ represents the number of cases, $I_{(t)}$ represents the number of the *I* population fitted by the model. A smaller MAPE value indicates a closer fit between the model results and the actual data, indicating a higher prediction accuracy of the model.

$$MAPE = \frac{100\%}{t} \sum_{i=1}^{t} \left| \frac{R_{(t)} - I_{(t)}}{R_{(t)}} \right| \tag{21}$$

Actual data from Wuhan city when COVID-19 occurred are used for the model verification. According to the data from the Wuhan Municipal Health Commission, this paper takes the initial values of the SEIR model as shown in Table 1 to simulate Formula (3) in MATLAB software (R2021b). The simulated number of the *I* population is then fitted with the actual infection data from Wuhan for 35 days, from 23 January 2020 to 26 February 2020. At this point, the MAPE is 18.08%. The fitting curve is shown in Figure 5. This demonstrates that the SEIR epidemiological prediction model proposed in this paper can accurately predict the generation of medical waste.

**Table 1.** Initial values for the epidemiological dynamics model.

| Parameters | Parameter Description | Parameter Values |
|:---:|:---:|:---:|
| $S_{(0)}$ | Initial value of susceptible population | 110,810.00 |
| $E_{(0)}$ | Initial value of exposed population | 105.00 |
| $I_{(0)}$ | Initial value of infected population | 0.00 |
| $R_{(0)}$ | Initial value of removed population | 0.00 |
| $N$ | Total population potentially affected | 110,915.00 |
| $\beta$ | Infection rate from exposed to susceptible | $1987.59 \times 10^{-4}$ |
| $\delta$ | Infection rate from infected to susceptible | $4802.70 \times 10^{-4}$ |
| $\mu$ | Recovery rate | $585.11 \times 10^{-4}$ |
| $\gamma$ | Death rate | $12.70 \times 10^{-4}$ |
| $\alpha$ | Conversion rate from exposed to infected | $1228.57 \times 10^{-4}$ |

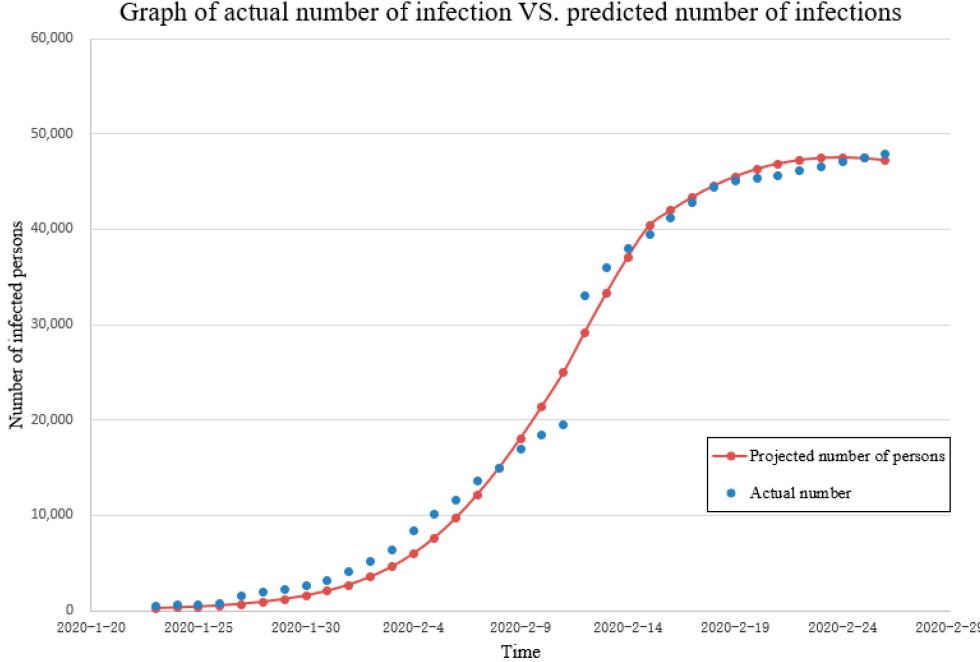

**Figure 5.** Comparison between predicted values and actual confirmed patient numbers.

*3.2. Case Setup and Solution Results*

The relevant data used in the case analysis are sourced from the Xi'an Statistical Yearbook and the Xi'an Health Information website, with parameters for processing centers referenced from the literature [21]. In this case, 79 designated hospitals and 11 isolation points, as government-designated collection points, are considered. Additionally, 5 existing waste processing centers—Senxin Processing Center ($j = 1$), Taida Processing Center ($j = 2$), Jiangcungou Processing Center ($j = 3$), Lantian Processing Center ($j = 4$), and Weida Processing Center ($j = 5$)—are selected as candidate points for waste treatment during the epidemic, with specific locations shown in Figure 6. Formula (22) is used to convert the latitude and longitude distances between nodes into actual distances $l_{ij}$ between node $i$ and node $j$. Here, the Earth's radius $R$ is 6371 km, and $(x_1, y_1)$ and $(x_2, y_2)$ are the latitude and longitude coordinates of the nodes in question. In addition, the per capita economic loss caused by the leakage of medical waste is 34275 CNY [4]. The values of other parameters can be found in Tables 2 and 3.

$$l_{ij} = Rarccos(\sin y_1 \sin y_2 + \cos y_1 \cos y_2 \cos(x_1 - x_2)) \tag{22}$$

**Table 2.** Parameters setting for processing centers.

| Parameters | $j = 1$ | $j = 2$ | $j = 3$ | $j = 4$ | $j = 5$ |
|---|---|---|---|---|---|
| $Hd_j$ | 45.00 | 45.00 | 45.00 | 45.00 | 45.00 |
| $Od_j$ | 5.52 | 5.83 | 5.27 | 5.65 | 5.43 |
| $Sd_j$ | 1061.85 | 889.76 | 904.77 | 1029.43 | 935.19 |
| $\rho d_j$ | 12.74 | 10.67 | 10.85 | 12.35 | 11.22 |
| $Bd_j$ | 2175.00 | 2062.50 | 2250 | 2100 | 2212.5 |

**Table 3.** Relevant parameters for transfer vehicles.

| $Ok$ | $V$ | $Hk$ | $r$ | $Bk$ |
|---|---|---|---|---|
| 400.00 | 60.00 | 15.00 | 0.50 | 2.50 |

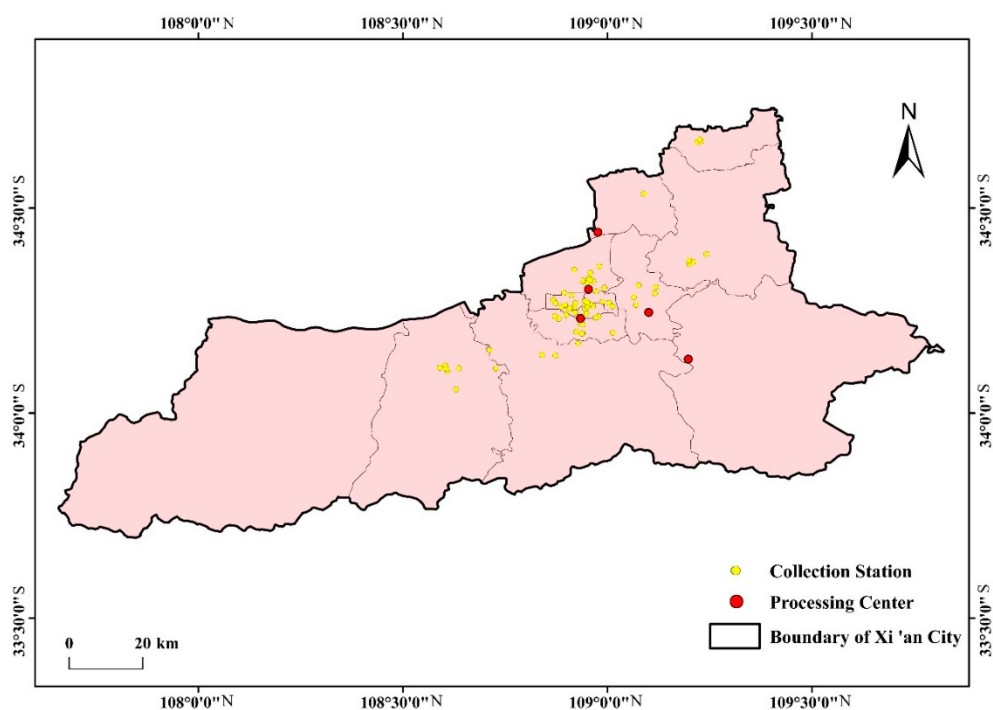

**Figure 6.** Distribution map of candidate points for processing centers and collection points.

The numerical experiments in this paper are conducted on a system with an Intel Core i7-9750H CPU @2.60 GHz, 8 GB of memory, and Windows 10 operating system using MATLAB R2021b. The parameters for the NSGA-II algorithm are set as follows: local search iteration is 800, *NP* (population size) is 100, crossover probability (*Pc*) is 0.9, and mutation probability (*Pm*) is to 0.1. The optimal results are selected through 30 random tests.

The solutions obtained by the NSGA-II algorithm are shown in Table 4. Here, opt 1 and opt 2 represent the solutions obtained under the optimal conditions for economic cost ($F_1$) and risk ($F_2$) individually. In addition, opt 3 is the optimal non-dominated solution with maximum crowding distance. The diagrams of the solutions are shown in Figure 7.

**Table 4.** Solution set of optimal values for each objective.

| Solution | Processing Centers | Objective Functions | |
|---|---|---|---|
| | | $F_1$ (CNY) | $F_2$ (CNY $\times$ ton) |
| opt 1 | $j = 1, 2$ | 250,916.13 | 85,396.41 |
| opt 2 | $j = 2, 3$ | 264,211.54 | 74,749.43 |
| opt 3 | $j = 2, 3$ | 263,314.26 | 76,981.22 |

Analyzing the obtained Pareto solution from different perspectives of decision makers, the results reveal the following.

(1) From the government's perspective, opt 3 shown in Figure 7c is more balanced in terms of economic cost and risk. In terms of risk, it is 11.1% less than opt 1, while only 2.91% more than opt 2. In terms of economic cost, it is 0.34% less than opt 2, while only 4.71% more than opt 1.

(2) From the perspective of the processing centers, opt 1 has the lowest economic cost, 5.30% less than opt 2, and 4.70% less than opt 3, making it the optimal choice.

(3) Considering public risk, opt 2 is the best choice, as shown in Figure 7b. The corresponding risk is 14.24% less than opt 1 and 2.99% less than opt 3.

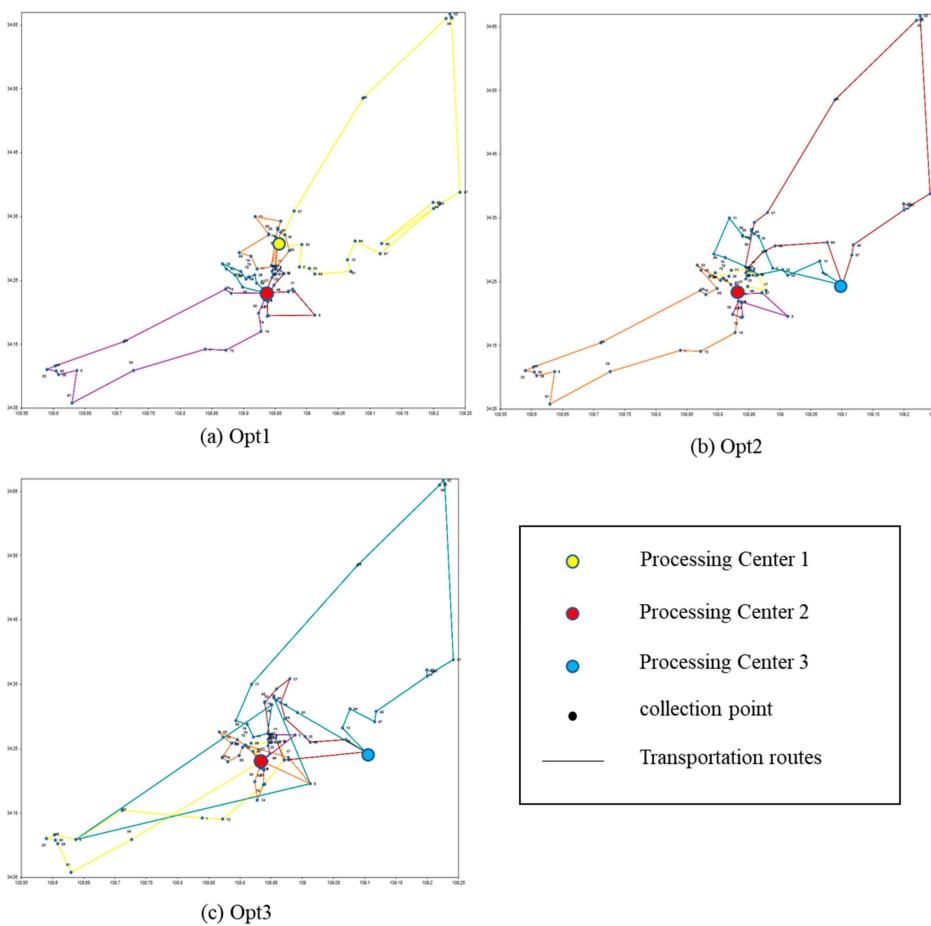

**Figure 7.** Schematic diagram of location-routing option for each objective.

*3.3. Analysis of Disposal Demand in Different Scenarios*

In order to study the variation in costs and risks of the reverse logistics network for medical waste in two scenarios of a routine and public health emergency, a comparison is now being made between the solutions for both scenarios, as shown in Table 5.

**Table 5.** Comparison of decision solutions in different scenarios.

| Different Scenarios | $W_{(t)}$ (ton) | $P_{(t)}$ | Processing Centers | Number of Transfer Vehicles | $F_1$ (CNY) | $F_2$ (CNY $\times$ ton) |
|---|---|---|---|---|---|---|
| Routine Scenario | 20.25 | 0.21 | $j = 3$ | 2 | 99,894.11 | 11,274.97 |
| Public Health Emergency Scenario | 65.61 | 0.86 | $j = 1, 2$ | 5 | 250,929.17 | 85,679.76 |

By analyzing the above results, it can be observed that compared to routine scenarios, when a public health emergency occurs, the objective functions significantly increase. Specifically, the economic cost rises by 151.19%, and risk increases by 659.91%. Meanwhile, the required number of processing centers increases by 100%, and the number of transfer vehicles increases by 150%. This indicates that the existing infrastructure is far from sufficient to meet the processing capacity needed during a public health emergency.

*3.4. Analysis of Different Risk Magnitudes*

To adjust the decisions as the magnitude of a public health emergency changes, this paper designs cases for large-, medium-, and small-scale public health emergencies following the classification rules for emergency in the "Emergency Response Law of the People's Republic of China". The results are shown in Table 6.

**Table 6.** Results of solutions under different public health emergency scales.

| Different Scales | $I_{(t)}$ | $W_{(t)}$ (ton) | $P_{(t)}$ | Processing Centers | Number of Transfer Vehicles | $F_1$ (CNY) | $F_2$ (CNY × ton) |
|---|---|---|---|---|---|---|---|
| Small Scale | 596 | 25.99 | 0.29 | $j = 3$ | 2 | 112,841.93 | 16,220.09 |
| Medium Scale | 1514 | 34.53 | 0.39 | $j = 2$ | 3 | 131,549.33 | 21,146.57 |
| Large Scale | 4569 | 65.61 | 0.86 | $j = 1, 2$ | 5 | 250,929.17 | 85,679.76 |

From Table 6, it is evident that with the change in the scale of a public health emergency, the number of people affected increases significantly. Correspondingly, the generation of medical waste also increases. Without timely adjustments to the treatment plans, the existing plans cannot meet the sudden surge in processing requirements. Decision makers may need to increase the construction of appropriate processing centers and dispatch more transfer vehicles to meet the disposal demand. Therefore, the model proposed in this paper can provide decision makers with reasonable facility location and transportation path solutions based on the different stages of the development of public health emergencies.

*3.5. Analysis of Transfer Vehicle Selection*

Considering that the capacity of transfer vehicles directly affects the transportation efficiency, this paper conducts a sensitivity analysis of vehicle capacity to study the impact of vehicle selection on the objective functions under different scenarios. The decision results for vehicle capacities of 5, 10, 15, and 20 tons are shown in Table 7.

**Table 7.** Impact of different vehicle capacities on objective functions under different scenarios.

| Vehicle Capacity | Routine Scenario | | | Public Health Emergency | | |
|---|---|---|---|---|---|---|
| | Number of Vehicles | $F_1$ (CNY) | $F_2$ (CNY × ton) | Number of Vehicles | $F_1$ (CNY) | $F_2$ (CNY × ton) |
| 5 | 5 | 99,451.09 | 12,909.09 | 14 | 25,0745.40 | 97,779.90 |
| 10 | 3 | 99,641.70 | 12,012.22 | 7 | 25,0861.50 | 93,482.94 |
| 15 | 2 | 99,973.10 | 12,586.80 | 5 | 25,1419.70 | 87,475.72 |
| 20 | 2 | 102,659.80 | 12,734.68 | 4 | 25,2242.00 | 88,765.20 |

From Table 7, it can be observed that under routine scenarios, as the vehicle capacity increases from 5 to 20 tons, the risk initially decreases and then increases. When the vehicle capacity is 10 tons, the risk is minimized. In the case of a public health emergency, the risk shows a similar trend of initially decreasing and then increasing, with the minimum risk occurring when the vehicle capacity is 15 tons. Therefore, it can be concluded that under routine scenarios, choosing transfer vehicles with a capacity of 10 tons can gain better benefits. In a public health emergency, transfer vehicles with a larger capacity, such as 15 tons, should be chosen to enhance transportation efficiency and reduce transportation risks.

**4. Conclusions**

This paper explores the optimization of a reverse logistics network for medical waste under an uncertain proposal demand. To address the issue of unstable demand for medical waste disposal, this study has developed a prediction model based on the SEIR epidemiological dynamics method. The model forecasts the disposal needs for medical waste in both routine and public health emergency scenarios. Additionally, a medical waste reverse logistics network model has been established, taking into account both cost and risk factors. The NSGA-II algorithm is designed to solve this model. Finally, the effectiveness of the model and algorithm has been validated through real cases in Xi'an city and multiple test cases.

The research results can provide decision support for relevant stakeholders in determining the optimal facility location and transportation routes under different scenarios. Based on the computational results, the main conclusions of this study are as follows.

(1) The medical waste prediction model designed in this study can reasonably and effectively predict the amount of medical waste generated by fitting the number of infections in Wuhan from 23 January 2020 to 26 February 2020, with a mean relative error (MAPE) of 18.08%.

(2) The medical waste reverse logistics network model proposed in this paper can provide corresponding location and transportation route solutions from the perspectives of different stakeholders such as the government, processing centers, and the public. In a public health emergency, decision makers can formulate reasonable plans to increase processing facilities' transportation plans based on the scale of the public health emergency.

(3) A sensitivity analysis of vehicle capacity indicates that, under routine scenarios, a 10-ton capacity transfer vehicle is preferable. For public health emergencies, on the other hand, a transfer vehicle with a capacity of 15 tons should be selected. This approach helps reduce transportation risks and economic costs.

## 5. Future and Prospects

(1) With the increasing maturity of AI technology, we believe that AI technology can be used to predict the amount and type of medical waste generated using data analysis and machine learning algorithms; it can also be used in combination with GIS technology to provide decision-making support for the siting of medical waste treatment centers; and the use of real-time traffic data for real-time optimization of the transfer paths of medical waste, among other applications to solve the problem of medical waste treatment.

(2) Due to the limitation of this paper, only data from China were collected for the study, and in the future, data from other countries will need to be considered to validate the model mentioned in this paper.

(3) With the increasing burden of road safety, aspects such as noise and exhaust pollutants should also be considered in future studies to minimize pollution.

(4) Since the siting of temporary hospitals has an important impact on the design of medical waste reverse logistics networks, the location of waste treatment centers, reverse logistics path planning, and emergency response capabilities need to be fully considered in the siting process to ensure that waste can be handled and transferred in a safe and timely manner. Therefore, the location of temporary hospitals should also be considered in future studies.

**Author Contributions:** L.Z.: conceptualization, methodology, investigation, resources, writing—review and editing. T.D.: conceptualization, software, formal analysis, data curation, writing—original draft preparation. Z.L.: methodology, validation, resources, visualization, supervision. All authors have read and agreed to the published version of the manuscript.

**Funding:** This research was funded by the Natural Science Foundation of Shaanxi Province of China, grant number 2023-JC-QN-0796 and Humanities and Social Sciences Fund of the Ministry of Education, grant number 22YJCZH274.

**Institutional Review Board Statement:** Not applicable.

**Informed Consent Statement:** Not applicable.

**Data Availability Statement:** Data are contained within the article.

**Conflicts of Interest:** The authors declare no conflicts of interest.

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
