# Peer review of "Reverse Logistics Network Design for Medical Waste Disposal under the Scenario of Uncertain Proposal Demand"

_sustainability, doi:10.3390/su16072996_

Round 1

Reviewer 1 Report

Comments and Suggestions for Authors

Apart form changing the term "scholars" to experts or scientists, I have no objections to this paper. It is well written, simple enough in concept and seems that it can be applicated to solving of real problems.

There is one thing that I would like to see... How would AI solve this problem? Some comparison would be nice to see.

Comments on the Quality of English Language

Maybe some proof reading (English) would be needed.

Author Response

Dear Reviewer:

Thank you for your valuable comments concerning our manuscript entitled “Reverse Logistics Network Design for Medical Waste under the Scenario of Uncertain Proposal Demand” (Manuscript ID: sustainability-2904309). The comments are helpful for revising and improving our paper. We have considered the comments carefully and tried our best to make improvement. Revised portion are marked in red in the revised paper. The list of main actions and the response to your comments are as flowing:

Responses to Reviewer

Comment 1:

Apart form changing the term "scholars" to experts or scientists, I have no objections to this paper. It is well written, simple enough in concept and seems that it can be applicated to solving of real problems.

Response:

Thanks to your reminder, we have corrected the word "scholars" in the revised manuscript.

Comment 2:

There is one thing that I would like to see... How would AI solve this problem? Some comparison would be nice to see.

Response:

Based on the reviewers' suggestions, we have added a new section 5 (Future and Prospects) to the revised manuscript to discuss the role that AI will play in the design of healthcare waste reverse logistics networks.

Supplementary Specifications:

Except for the above-mentioned revisions, some extra revisions requested by other reviewers are made in the manuscript.

We appreciate for the editor and reviewer’ s work earnestly again, and we hope that the revise will meet with your approval.

Reviewer 2 Report

Comments and Suggestions for Authors

Paper Title: Reverse Logistics Network Design for Medical Waste under the Scenario of Uncertain Proposal Demand 

The present research examines the reverse logistics network optimization for medical waste under uncertain demand. They prepare a prediction model of medical waste based on the Susceptible cases of the disease, infectious cases of the disease, and recovered cases of the disease (SEIR) considering routine and public health emergency scenarios. Later on, they propose A bi-objective location-routing optimization model for a medical waste reverse logistics network by optimizing the total economic cost and potential risk throughout the entire logistics process.

Research Contribution:

They employ a nondominated sorting genetic algorithm, (NSGA-II) algorithm for different situations of routine and emergency public health cases in Xi’an city on large, medium, and small scales.

 Writing issues:

1)      The abstract and the introduction have the same beginning, hence may be modified.

2)      The title may be modified as “Reverse Logistics Network Design for Medical Waste Disposal under the Scenario of Uncertain Proposal Demand’ or on similar lines.

3)      Authors may clearly distinguish between equation numbers and numbers used in Figure 4 of The problem encoding diagram.

4)      Step 3 equation for “solution set { F1,F2,Fi},” may be properly written.

5)      Figure 3. Scope of influence of processing center and transportation path. (a) ‘Process’ may be corrected.

 Technical issues:

1)      Please refer to “The epidemiological dynamics-based prediction model is …. medical waste with a Mean Absolute Percentage Error (MAPE) of 18.08%’. A MAPE value of <10% is preferred. The statement “The medical waste prediction model designed in this study can accurately forecast the generation of medical waste by…” may be modified accordingly.

2)      Please refer to “However, given that reverse …. cannot meet the current disposal needs for medical waste.” Authors may provide clarity on the type of “medical waste” considered that needs reverse logistics to justify the incapability of prevailing methods to create the research gap.

3)      Please refer to "where 1-3 represent processing centers ' 1 and 2 are not shown in Figure 4 for the problem encoding diagram.

4)      Legend is missing for Figure 5. Comparison between Predicted Values and Actual Confirmed Patient Numbers.

5)      Please refer to Figure 7. Schematic Diagram of Location-Routing option for Each Objective., The authors need to provide more explanation and clarity. What is the impact of crowding distance on the Location-Routing option?

6)      Authors may clarify how risk is quantified, as mentioned “and risk increases by 659.91%.’

 Conclusion

 1)      The implication of “vehicles with larger capacity should be chosen.”

2)      Effect on Transportation expenses and complexity due to increased burden on road safety, pollution in the area of noise and exhaust pollutants.

3)      Future research direction to minimize effects.

Comments on the Quality of English Language

The manuscript has some typos and grammatical mistakes, hence careful editing is needed.  

1)   Please refer to “With the development of healthcare industry,…..”  should be “With the development of the healthcare industry,…..” 

2)   Please refer to “…in the event of public health emergency…” should be “…in the event of a public health emergency…”.

3)   "From a operational perspective..." should be "From an operational perspective..."

4)   "..specific waste are relatively fixed,..." should be "..specific waste is relatively fixed,...".

5)   "They conducted sensitivity analysis.." should be "They conducted a sensitivity analysis.."

6)   "..on the other hand, focus on ..." should be "..on the other hand, focuses on ..."

7)   "...advantages lies in: Firstly, reduce ...' should be "...advantages lie in the: Firstly,...'

8)   “…employ an elitist strategy…” should be “…employs an elitist strategy…”

9)   “…it can be observed that compared to routine scenario,” should be “…it can be observed that compared to routine scenarios,”

10)  "...a sensitivity analysis on vehicle capacity ..." should be "...a sensitivity analysis of vehicle capacity ..."

11)  "it can be concluded that under routine scenario, " should be "it can be concluded that under routine scenarios, "

Author Response

Dear Reviewer:

Thank you for your valuable comments concerning our manuscript entitled “Reverse Logistics Network Design for Medical Waste under the Scenario of Uncertain Proposal Demand” (Manuscript ID: sustainability-2904309). The comments are helpful for revising and improving our paper. We have considered the comments carefully and tried our best to make improvement. Revised portion are marked in red in the revised paper. The list of main actions and the response to your comments are as flowing:

Responses to Reviewer

Writing issues:

Comment 1:

The abstract and the introduction have the same beginning, hence may be modified.

Response:

We think your suggestions for the content of the article's opening and summary sections are excellent. We have made some adjustments to the introductory part of the article.

Comment 2:

The title may be modified as “Reverse Logistics Network Design for Medical Waste Disposal under the Scenario of Uncertain Proposal Demand’ or on similar lines.

Response:

After discussion, we agreed that adding the word "Disposal" to the title was a very good suggestion.

Comment 3:

Authors may clearly distinguish between equation numbers and numbers used in Figure 4 of the problem encoding diagram.

Response:

Thank you very much for your suggestion regarding the error in the figures in Figure 4, which has been corrected in the paper.

Comment 4:

Step 3 equation for “solution set { F1,F2,…Fi},” may be properly written.

Response:

We have adjusted the way the set { F1, F2, ..., Fi} is written according to your suggestion.

Comment 5:

Figure 3. Scope of influence of processing center and transportation path. (a) ‘Process’ may be corrected.

Response:

Thanks to your reminder, we have corrected the spelling error in Figure 3 by replacing the word "process" with "processing".

Technical issues:

Comment 1:

Please refer to “The epidemiological dynamics-based prediction model is …. medical waste with a Mean Absolute Percentage Error (MAPE) of 18.08%’. A MAPE value of <10% is preferred. The statement “The medical waste prediction model designed in this study can accurately forecast the generation of medical waste by…” may be modified accordingly.

Response:

We have modified the wording based on actual results and will continue to improve the longitude of the model in subsequent studies.

Comment 2:

Please refer to “However, given that reverse …. cannot meet the current disposal needs for medical waste.” Authors may provide clarity on the type of “medical waste” considered that needs reverse logistics to justify the incapability of prevailing methods to create the research gap.

Response:

Thank you very much for your suggestion, we think it is a good one. We have included the type of medical waste studied in this paper at the end of the second paragraph of the introduction section, which enables us to emphasize the specificity of this type of medical waste in order to highlight the contribution of our study.

Comment 3:

Please refer to "where 1-3 represent processing centers ' 1 and 2 are not shown in Figure 4 for the problem encoding diagram.

Response:

As shown in Figure 4, processing centers that are not selected in the feasible scenario coding will not be shown in the coding map. Processing center 3 is operational; since processing centers 1 and 2 were not selected, they will not be shown in the coding diagram and will not be operational.

Comment 4:

Legend is missing for Figure 5. Comparison between Predicted Values and Actual Confirmed Patient Numbers.

Response:

You pointed out the lack of a legend in Figure 5. We have carefully considered your suggestion and have revised and adjusted Figure 5 accordingly to supplement the missing legend. The new legend will provide the reader with a clearer understanding of the data in the figure and ensure the accuracy and completeness of the chart.

Comment 5:

Please refer to Figure 7. Schematic Diagram of Location-Routing option for Each Objective., The authors need to provide more explanation and clarity. What is the impact of crowding distance on the Location-Routing option?

Response:

We have adjusted Figure 7 to make it as clear and easy to understand as possible. We have also added corresponding schematic diagrams to provide readers with more comprehensive information. These improvements will ensure the accuracy and clarity of the diagrams and enhance the quality of the paper.

In NSGA-II, the crowding distance is used to characterize the degree of similarity of a given feasible solution to other solutions. A lower crowding distance indicates a higher degree of similarity of solutions; a higher crowding distance indicates a richer diversity of solutions. Therefore, the impact of the choice of solutions with different congestion distances on the site-path selection can be interpreted as whether the solution fits our choice. Usually we choose the solution with smaller crowding distance.

Comment 6:

Authors may clarify how risk is quantified, as mentioned “and risk increases by 659.91%.’

Response:

The part of the risk quantification you mentioned is explained in detail below:

(1) In Subsection 2.3, we have introduced formulas (6) and (7) to quantify the risks of the processing centers and transportation routes. These formulas are based on previous studies and practical experience and take into account various factors, including the geographical location of the facilities, the number of people in the surrounding areas who may be affected, the degree of impact, and so on, in order to assess the risks that may be faced by the processing centers and transportation routes. Through these formulas, we can quantitatively measure and compare the risk levels of different processing centers and transportation routes, thereby providing a reference basis for decision-making.

(2) In addition, in Equation (8), we quantify more specifically the quantitative relationship between the probability of risk and the amount of clinical waste generated. This formula takes into account the correlation between the probability of medical waste generation and the amount of waste generated. The introduction and application of these formulas are intended to help us understand and assess more comprehensively the various risks that may be faced in the treatment of medical waste, so that effective measures can be taken for risk management and control.

(3) In section 3.3 of the paper, we compared the solution results under two scenarios, namely, daily situation and public health emergency, by subtracting the F2 of daily situation from the F2 of public health emergency and dividing it by the F2 of daily situation. the results found that the risk of public health emergency is 659.91% higher than that of daily situation.

Conclusion:

Comment 1:

The implication of “vehicles with larger capacity should be chosen.”

Response:

In response to your suggestions, we have revised the text accordingly and have included a more specific and precise statement in the conclusion. In the original text, we used the phrase "vehicles with larger displacement should be selected" to summarize the recommendation to select vehicles with larger displacement. However, with your guidance, we realized that we needed to be more specific about our recommendations for selecting vehicles in public health emergencies. Therefore, we have revised the statement in the original text to read "In public health emergencies, a transfer vehicle with a 15-ton capacity should be selected" in order to express our recommendation more clearly.

Comment 2:

Effect on Transportation expenses and complexity due to increased burden on road safety, pollution in the area of noise and exhaust pollutants.

Future research direction to minimize effects.

Response:

Thank you for your suggestions on the future research direction of this paper, the increased burden of road safety, noise and exhaust pollution, which are important to guide the direction of our team's work. Your suggestions provide a new perspective for us to think deeply about future research directions, especially in the field of road safety and environmental pollution. Our team will pay close attention to your suggestions and actively adjust and explore them in our future work, with a view to being able to better respond to the concerns of the society and the scientific community and make greater contributions to solving related problems.

Comments on the Quality of English Language

Comment :

The manuscript has some typos and grammatical mistakes, hence careful editing is needed.

Response:

We have done our best to touch up the language in the revised draft based on your suggestions.

Supplementary Specifications:

Except for the above-mentioned revisions, some extra revisions requested by other reviewers are made in the manuscript.

We appreciate for the editor and reviewer’ s work earnestly again, and we hope that the revise will meet with your approval.

Reviewer 3 Report

Comments and Suggestions for Authors

The article addresses the issue of reverse logistics by addressing the problem of medical waste. Such a topic has gained in importance, however also in popularity since 2020 due to the COVID pandemic. A similar problem is discussed in Kargar, S., Pourmehdi, M., & Paydar, M. M. (2020). Reverse logistics network design for medical waste management in the epidemic outbreak of the novel coronavirus (COVID-19). In Science of The Total Environment (Vol. 746, p. 141183). Elsevier BV. https://doi.org/10.1016/j.scitotenv.2020.141183 and in other articles cited by the authors (Govindan, 2022). The purpose of the article is clearly explained. Similar recent studies have been presented. The medical waste prediction model, based on the SEIR epidemiological dynamics method, is the most interesting part of the study.  The location and route optimization model for the medical waste reverse logistics network has well-defined objectives and constraints. Verification of the model on historical data has proven its effectiveness. It was performed only for China, so it would be good to refer to other countries. Also, the problem of locating temporary hospitals or temporary waste centres was not considered. In the conclusions, the authors did not show the direction of further work.

Author Response

Dear Reviewer:

Thank you for your valuable comments concerning our manuscript entitled “Reverse Logistics Network Design for Medical Waste under the Scenario of Uncertain Proposal Demand” (Manuscript ID: sustainability-2904309). The comments are helpful for revising and improving our paper. We have considered the comments carefully and tried our best to make improvement. Revised portion are marked in red in the revised paper. The list of main actions and the response to your comments are as flowing:

Responses to Reviewer

Comment 1:

A similar problem is discussed in Kargar, S., Pourmehdi, M., & Paydar, M. M. (2020). Reverse logistics network design for medical waste management in the epidemic outbreak of the novel coronavirus (COVID-19). In Science of The Total Environment (Vol. 746, p. 141183). Elsevier BV. https://doi.org/10.1016/j.scitotenv.2020.141183 and in other articles cited by the authors (Govindan, 2022).

Response:

We have scrutinized the literature and added the literature you suggested in the introduction section of the revised manuscript [3]. By adding this literature, we believe that our paper will be more complete and persuasive, and can better support our research ideas and conclusions.

Comment 2:

It was performed only for China, so it would be good to refer to other countries. In the conclusions, the authors did not show the direction of further work.

Response:

The issue you raised about the lack of data from other countries is indeed a limitation of our work. We fully recognize this and sincerely apologize to you for our failure to provide more comprehensive data. We will pay close attention to the suggestions you have made and take active steps in our future work to remedy the limitations of our study. We will start collecting and analyzing more data from other countries to make our study more global and universal.

Comment 3:

Also, the problem of locating temporary hospitals or temporary waste centres was not considered.

Response:

The medical waste reverse logistics network we study is a siting path problem, with three main components including medical waste prediction, siting of treatment centers, and optimization of transfer paths. For the time being, we have not considered the siting of the temporary hospital you mentioned. The suggestions you have made are of great significance in guiding our work, especially the suggestion that we should consider the siting of temporary hospitals. This points out the direction for our future work and helps us to conduct scientific research more rigorously.

Supplementary Specifications:

Except for the above-mentioned revisions, some extra revisions requested by other reviewers are made in the manuscript.

We appreciate for the editor and reviewer’ s work earnestly again, and we hope that the revise will meet with your approval.